# FS-Mol: A Few-Shot Learning Dataset of Molecules

**Megan Stanley**
Microsoft Research

**John Bronskill**
Microsoft Research
University of Cambridge

**Krzysztof Maziarz**
Microsoft Research

**Hubert Misztela**
Novartis

**Jessica Lanini**
Novartis

**Marwin Segler**
Microsoft Research

**Nadine Schneider**
Novartis

**Marc Brockschmidt**
Microsoft Research

## Abstract

Small datasets are ubiquitous in drug discovery as data generation is expensive and can be restricted for ethical reasons (e.g. in vivo experiments). A widely applied technique in early drug discovery to identify novel active molecules against a protein target is modeling quantitative structure-activity relationships (QSAR). It is known to be extremely challenging, as available measurements of compound activities range in the low dozens or hundreds. However, many such related datasets exist, each with a small number of datapoints, opening up the opportunity for few-shot learning after pretraining on a substantially larger corpus of data. At the same time, many few-shot learning methods are currently evaluated in the computer-vision domain. We propose that expansion into a new application, as well as the possibility to use explicitly graph-structured data, will drive exciting progress in few-shot learning. Here, we provide a few-shot learning dataset (FS-Mol) and complementary benchmarking procedure. We define a set of tasks on which few-shot learning methods can be evaluated, with a separate set of tasks for use in pretraining. In addition, we implement and evaluate a number of existing single-task, multi-task, and meta-learning approaches as baselines for the community. We hope that our dataset, support code release, and baselines will encourage future work on this extremely challenging new domain for few-shot learning.

## 1 Introduction

Deep Learning has led to tremendous progress in a variety of domains, such as computer vision [6], natural language processing [11], molecular design [49], chemical synthesis planning [45], and most recently protein folding [26]. In all of these cases, progress has been driven by large single-task datasets (e.g., 14M images in ImageNet; ∼500B tokens for GPT-3), often enabled by a combination of substantial manual labeling together with elegant methods able to learn from unlabeled data.

Computer-aided drug discovery has long made use of machine learning to predict the biological activity of molecules [33, 52, 56]. Such models are often most useful in the hit-to-lead and early lead optimization phase of a drug discovery project. After the biological target (often a protein) has been identified, initial hit molecules, which modulate the target but do not have yet have all required properties, need to be optimized towards a drug candidate suitable for clinical testing. This optimization phase proceeds by the time-consuming and expensive iterative procedure of designing new molecules, synthesizing them, and measuring their properties. Prediction using ML models reduces the number of molecules requiring synthesis and wet-lab measurement, considerably reducing the cost of drug discovery [44]. In the typical life cycle of a drug discovery project, only a few hundred molecules can be made and tested. With such small datasets, current deep learning approaches are in practice often not more effective than support vector machines, random forests or gradient-boosted tree models based on molecular feature representations, which have been tuned over decades [59].

35th Conference on Neural Information Processing Systems (NeurIPS 2021) Track on Datasets and Benchmarks.

However, data is gathered across many different targets (i.e. tasks) and different drug discovery projects, suggesting the possibility of knowledge transfer between low-data tasks. A common framework for such settings is few-shot learning [10, 18, 41, 53, 58, 64]. For example, meta-learning [18, 21, 53, 58] refers to algorithms which learn to learn; given an unseen task, such methods utilize their prior knowledge from previous training episodes to generalize and make predictions in a few-shot manner. The efficacy of this paradigm for enhancing molecular property prediction in low-data regimes has shown some promise [7, 9, 35, 36], but we note that there is no dataset specifically designed to benchmark few-shot learning methods for molecules and thus allow easy model comparison. In particular, this would require a dataset that provides both a large range of diverse tasks on which to perform an initial pretraining step as well as a set of low-data tasks on which to evaluate, with both of these chosen to be similar in nature.

From a machine learning point of view, few-shot learning has been primarily focused on the computer vision domain [29, 55, 58]. However, state-of-the-art performance has become relatively saturated [12, 24, 37, 54], and many approaches profit from pretraining on rich, highly related large datasets such as ImageNet [43], which are not available in other few-shot domains. To drive further algorithmic innovation and solutions to real-world scientific problems, we propose that few-shot learning should expand into alternate domains of applicability.

With our new dataset, we aim to inspire the development of machine learning methods that generalize well across a diverse distribution of tasks and adapt efficiently to minimal new data. We also consider it essential that models developed using such a dataset and its benchmarks should have value in real-world applications, and hence we propose to use Quantitative Structure-Activity Relationships (QSAR) as the data domain, in which the task is to predict the activity in inhibition or activation of a specific target protein given the structure of a molecule.

In this work, we make three key contributions:

- A Few-Shot Learning Dataset of Molecules (henceforth abbreviated to FS-Mol) that can demonstrate the utility of few-shot learning methods in an important domain, namely QSAR, which does not provide an obvious generic pretraining corpus (such as in NLP or computer vision). The proposed dataset is specifically designed to replicate the challenges of machine learning in the very low data regime of drug-discovery projects.

- A fixed benchmarking procedure on this dataset that allows to easily compare new few-shot learning methods in an apples-to-apples scenario. We hope this will encourage the development of novel methods for few-shot learning on structured data.

- The establishment of baselines against the new benchmark with representative few-shot learning methods.

## 2 Motivation

Our main goal is to provide a dataset, benchmarks and baselines to encourage development of machine learning methods for realistic drug discovery tasks. We here focus on the challenge of a novel QSAR task at the very early stages of a lead optimization project, where there there is only very little available data. This will enable the community to explore the effectiveness of different machine learning techniques in this domain, and develop new methods to meet the unique challenges of the data. Concretely, we are interested in helping to answer two core research questions:

**(RQ1)** Does knowledge learned from a large dataset transfer gainfully to previously unseen tasks?

**(RQ2)** How does the performance vary with the size of the training data available in a new task?

### 2.1 Background: Few-Shot Learning

Standard supervised single-task methods rely on one dataset $\mathcal{D}$ to train a model, and evaluation proceeds by examining the predictions made on a held-out test set. Such approaches are known to work well with large sets of training examples drawn from the same distribution as the test set. In the few-shot learning scenario, no large training set exists. Instead, we aim to leverage an advantageous initialization (for example obtained from *pretraining* on related tasks), followed by training (*adaptation*) on very few datapoints from a previously unseen task.

The methods used during pretraining can fall into one of three categories. *Self-supervised* methods make use of a very large unlabelled dataset on which proxy tasks such as prediction of masked parts of the input are defined [16]. *Multitask* [13] and *meta-learning* methods assume the availability of a set of related supervised training tasks $\mathcal{D}_{train} = \{\mathcal{T}_t\}_{t=1}^K$. In general, each task $\mathcal{T}_t$ is composed of a *support set* $\mathcal{T}_{t,support}$, and a *query set* $\mathcal{T}_{t,query}$. The support set consists of examples with features $\mathbf{x}_i^t$ and labels $y_i^t$ which can be used to train a model, and the model is given the features $\mathbf{x}_j^t$, to predict the label $y_j^t$ of the query set.[1] At what we will term *few-shot testing* time, a model is given access to an unseen new task $\mathcal{T}_u$. It is then expected to predict labels on $\mathcal{T}_{u,query}$, while having access to the features and labels of $\mathcal{T}_{u,support}$. Consequently, evaluating these methods requires two disjoint sets of tasks: $\mathcal{D}_{train}$ (used for pretraining) and $\mathcal{D}_{test}$ (used to evaluate pre-trained models).

## 2.2 Desired Attributes of a QSAR Few-Shot Dataset and Benchmark

Given our research questions and the structure of most few-shot learning techniques, we can hence derive design requirements for our new dataset.

Concretely, to be amenable to few-shot learning, it should provide a large set $\mathcal{D}_{train}$ of training tasks useful for pretraining, and a disjoint set $\mathcal{D}_{test}$ of test tasks that are related to the training tasks. To enable an analysis of the extent of generalization to new tasks, $\mathcal{D}_{test}$ should contain both tasks that are very similar to the training data, as well as others that require more specialization at few-shot testing time. Finally, our test tasks should be chosen such that we can evaluate the amenability of evaluated methods to different support set sizes.

At the same time, we aim to construct a benchmark that is relevant for real-world drug-discovery projects. To this end, the number of samples for some tasks should be small, reflecting early-stage projects with access to measurements for fewer than 100 compounds. The considered molecules should be drug-like, and the tasks themselves should include a broad range of drug targets. Finally, the labels should be drawn from real measurements, to reflect the noise observed in wet-lab measurements for a novel target.

Overall, a benchmark on this dataset should capture the efficacy of the few-shot learning method in generalization to new tasks, i.e., does the adaptation method perform better than a single-task method exposed to the set $\mathcal{T}_{u,support}$ alone (answering **(RQ1)**), and can this adaptation use a minimal amount of data, answering **(RQ2)**.

## 3 The Few-Shot Learning Dataset of Molecules

We construct our dataset as a careful selection of data from ChEMBL27 [3]. We provide an overview in this section, but refer to our source code release (and in particular `ExtractDataset.ipynb` at `https://github.com/microsoft/FS-Mol/` for an exact, executable and reproducible definition.

**Selection of molecular property prediction tasks**    ChEMBL contains the results of many experiments, termed "assays", each having a unique experiment ID. We retained only those measurements referring to small molecule activity (IC50 or EC50) [48], and removed all compounds with a molecular weight $\geq 900$ Dalton in order to ensure only drug-like molecules are included. We then applied a standard cleaning and canonicalization procedure to all compounds (see details in our code release) and stored them as SMILES strings [61]. Assays were then selected to have at least 32 datapoints and not more than 5000 datapoints. The reason why we remove large assays is that they often come from high-throughput screens (HTS), and thus contain a high percentage of inactive compounds, and are very noisy, rendering the challenge more complex. We further exclude all assays that are not associated with a specific target protein ID. We view each selected, filtered assay as a single task in our few-shot learning dataset, as described in section 2.1. We only consider assays with a single protein target (where the same target may be the subject of several separate assays), and treat assays as separate tasks to avoid inter-assay noise often seen when combining measurements [27].

**Split into pretraining and test tasks**    As part of our dataset release, we provide a split into pretraining tasks $\mathcal{D}_{train}$ and (few-shot) test tasks $\mathcal{D}_{test}$. In order to derive disjoint task sets, we require that all selected assays are associated with a specific protein target. We avoid an overlap

---

[1]In the single-task scenario, the "support" set is called the training data, and the "query" set is the test data.

of very related tasks between pretraining and the test set by splitting tasks such that protein targets are used either only in the training or at most once in the test data.[2] We identified few-shot testing tasks from the subset of tasks that address enzyme targets. This choice enabled us to partition the set $\mathcal{D}_{test}$ by EC (Enzyme Commission) number [2] to permit sub-benchmarks within the overall benchmark set. The best few-shot learners are those able to perform well across all sub-benchmarks. Few-shot testing tasks were required to contain $> 128$ compounds to allow comparison of model adaptation performance across a range of available support set sizes, necessary to answer (**RQ2**). Our final $\mathcal{D}_{test}$ comprises 157 tasks, while $\mathcal{D}_{train}$ has 4938 tasks. We additionally also provide a $\mathcal{D}_{valid}$ set, consisting of 40 tasks selected in the same manner as $\mathcal{D}_{test}$, to aid the development of few-shot learning methods. In this way, our proposed meta-testing tasks closely mimic the new-lead optimization problem, where a completely unseen task is presented for adaptation.

We encourage the use of the set $\mathcal{D}_{train}$ not only for pretraining as described in section 2.1, but also as a task-specific phase following other large-data pretraining methods [23]. To enable this, the code implementing the dataset extraction and cleaning protocol outlined above can easily be adapted to select a larger set of assays than our chosen $\mathcal{D}_{train}$ (for example by also considering assays that are not annotated with a protein target).

**Binary Classification Task**  While the raw ChEMBL data provides activity as a floating point number, treating this as a regression target is known to be extremely hard (for reasons including measurement noise, narrow measurement range and that low/high values are often only encoded as a boundary constant). Instead, many practitioners only consider a binary classification task into active/inactive compounds, which is substantially more robust and often good enough to make decisions in the drug discovery process. While in practice thresholds for this would be defined on a by-project basis, we opt for an automated thresholding procedure based on the $IC_{50}$ or $EC_{50}$ value available for each compound. The median value over compounds in an assay defines the threshold, but the range of allowed thresholds are fixed to $5 \leq pXC \leq 7$ for enzyme targets and $4 \leq pXC \leq 6$ for all other protein targets, where $pXC = -\log_{10}(XC_{50})$. Should a median be found outside this range, a threshold $pXC = 5$ is applied, in keeping with fixed-threshold approaches taken elsewhere [31]. In this way, we ensure that label classes are more balanced to avoid further issues with highly imbalanced data diluting the comparison across different methods; assays where the median falls outside the prescribed range will be filtered if their classes are strongly imbalanced. These represent either very late stage optimization or very early or high-throughput screens. We include only those for which the active ratio falls between 0.30 and 0.70 (see Figure 1b).

**Dataset statistics**  FS-Mol consists of a total of 5120 separate assays, with 233,786 unique compounds. While assays address unique targets to prevent few-shot testing/pretraining overlap, many compounds are measured in multiple assays; $\mathcal{D}_{test}$ contains 27520 compounds, of which 15732 are unseen in $\mathcal{D}_{train}$ and $\mathcal{D}_{valid}$. The resulting task sizes are displayed in Figure 1a, where the mean number of compounds per task is 94, far below alternative datasets, reflecting the highly specific nature of the protein targets and the assays used to explore them.

**Features**  To encourage development of diverse approaches, our released dataset and supporting code provide three alternative featurization methods: (1) SMILES strings [61] for each compound, which may be used for NLP-inspired approaches [17, 25, 62] or to derive an arbitrary featurization; (2) Extended Connectivity Fingerprints (ECFP) and key molecular physical descriptors [42], which are standard choices in many machine learning approaches to QSAR; and (3) molecular graphs of atoms and bonds, to be used with methods such as graph neural networks.

## 4  Related Work

Given the long history of computer-assisted drug discovery, a substantial array of datasets is available for use in machine learning approaches to molecular property prediction. We summarize FS-Mol and related datasets in Table 1. Details on the various related datasets can be found in Appendix B.

We assess the available datasets according to their immediate "out of the box" suitability for the few-shot benchmarking scenario we are interested in.

---

[2]We note that it is possible, however, for the same target to be addressed by multiple assays, or tasks, within the pretraining set.

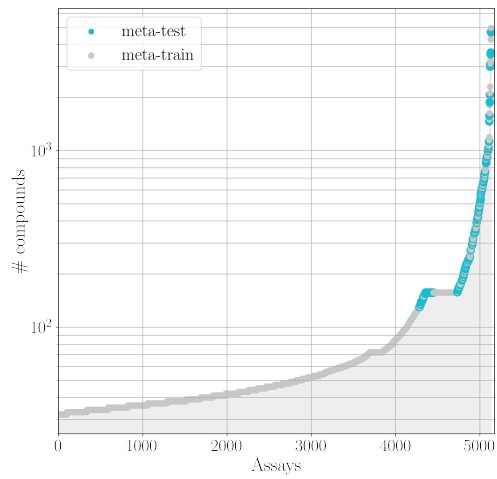 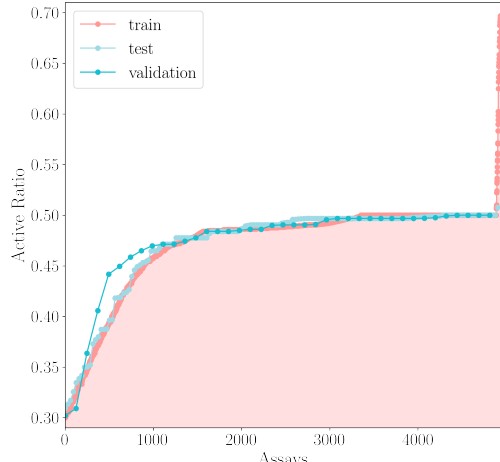

(a) Number of compounds in each assay. Meta-testing assays are drawn from larger assays to allow full support-set size comparison. Assays are ordered by size.

(b) The distribution of the proportion of active compounds in all sub-tasks in the dataset. Assays are ordered by proportion of active compounds.

Figure 1: FS-Mol statistics. Assays have sizes ranging from 32-5000, and most contain 30-50% active compounds.

Table 1: Key small molecule activity datasets for multiple tasks. All datasets are set as binary classification problems, although some additionally include underlying activity values (before thresholding).

|  | Datasets | | | |
|  | ExCAPE-ML | PCBA | LSC | FS-Mol |
| --- | --- | --- | --- | --- |
| # measurements | 49,316,517 | 34,017,170 | 5,100,411 | 489,133 |
| # compounds | 955,386 | 437,929 | 449,391 | 233,786 |
| # tasks | 526 | 128 | 1310 | 5120 |
| Mean # compounds / task | 93,758 | 265,759 | 3872 | 94 |
| Median # compounds / task | 1820 | 309,562 | 320 | 46 |
| Mean inactive:active / task | 268:1 | 46:1 | 7:1 | 1:1 |
| Raw values available? | Yes | No | No | Yes |
| Source | PubChem/ChEMBL | PubChem | ChEMBL18 | ChEMBL27 |

We require: (i) a large set of related QSAR prediction tasks to enable transfer to a diverse set of unseen prediction targets; (ii) a relatively small number of compounds per task to simulate the early lead optimization phase of a drug discovery project; (iii) well-balanced (i.e. active/inactive) classes, as this reflects a standard approach in QSAR modelling; (iv) a well-defined, context-aware $\{\mathcal{D}_{train}, \mathcal{D}_{test}\}$ task split. Examination of Table 1 reveals that FS-Mol is the only dataset that meets all these requirements. While FS-Mol contains fewer compounds and measurements that all the other compared datasets, it is the only one with a sufficiently large number of tasks where each contain relatively few compounds. We also note that our extraction pipeline enables easy addition of data as more becomes available in future editions of ChEMBL [3]. In addition, all FS-Mol tasks are well balanced by design and the $\{\mathcal{D}_{train}, \mathcal{D}_{test}\}$ split is explicit and fixed.

# 5 A Molecular Few-Shot Learning Benchmark

We now define a benchmarking methodology able to evaluate the utility of few-shot learning methods.

## 5.1 Evaluation Methodology

**Per-Task Dataset Splitting** In the setting of an early-stage drug discovery project, in which some leads (molecular structures demonstrating activity against the target) are known, models are used to evaluate variations of these leads and therefore choose promising development directions. Therefore temporal information regarding inclusion of a molecule in the dataset is the gold-standard to define dataset splits [46]. The data present in ChEMBL does not provide proper temporal information, and hence there is no inherent division of each task into support and query sets.

In the absence of temporal information, two options are regularly considered: (a) random splitting, using several different seeds to reduce the noise introduced by "lucky" splits, and (b) scaffold splitting, in which molecules with similar scaffolds (common, significant substructures) are grouped together, and groups are either entirely in the support or in the query set. The scaffold splitting scenario is clearly much harder and requires better generalization behavior of the considered models, but is also unrealistic in a real-world scenario of modeling project-specific data, where new molecules to be evaluated are variations of existing leads and usually share their scaffolds. Hence, we have decided to instead use repeated random splits for our evaluations.

Concretely, we perform ten-fold stratified random sampling for every task in $\mathcal{D}_{test}$ to create support and query sets. We perform this process for five support set sizes 16, 32, 64, 128, 256, in view of answering **(RQ2)**. Should there be insufficient overall samples to perform a split for a requested support set size, this evaluation point is passed over. The support sets can be used to train a single-task model, or to adapt a pre-trained model at few-shot adaptation time.

**Task-Level Metric** As core metric to evaluate considered models, we propose area under the precision-recall curve (AUPRC), which is sensitive to the class balance of our query sets. It also allows for a simple comparison to a trivial baseline: the AUPRC of a random classifier is equal to the percentage of positive points in the query set. To help answer **(RQ1)**, we use this observation to focus on the improvement of a given learned model $f$ over that trivial baseline, namely $\Delta$AUPRC $= \mathrm{AUPRC}(f(\mathcal{T}_{t,query})) - act_{t,query}/|\mathcal{T}_{t,query}|$, where $act_{t,query}$ refers to the number of active compounds in $\mathcal{T}_{t,query}$.

**Dataset-Level Evaluation** The task-level evaluation above can be extended to the entire set of test tasks, helping to reduce the noise stemming from particular tasks or "lucky" parameter choices. To this end, we consider the mean $\Delta$AUPRC at different support set sizes (allowing identification of models that are particularly well-suited to very small datasets) and the mean rank in comparison with other methods. Additionally, to reflect the differences in our considered tasks, we also break out evaluation results into smaller categories defined by the different EC numbers.

# 6 Experimental Baseline Evaluation

In this section, for reference for the community, we present results of standard methods on our new dataset to illustrate the potential and shortcomings of these techniques.

## 6.1 Baseline Methods

We provide a set of results for all three categories of few-shot learning, with representative methods of the use of this dataset in each. Model implementations used for the experiments are available in our source code release at `https://github.com/microsoft/FS-Mol`.

**Single-Task Methods** The reigning industry standard in in-silico modeling of very small tasks for drug discovery is the use of random forests (called RF below) and k-nearest neighbour models (kNN) on top of manually curated features (namely, extended connectivity fingerprints [42] and phys-chem descriptors [1]). To answer **(RQ1)**, we include these models in our few-shot learning evaluation. We used typical hyperparameter search configurations for these classes of models based on the extensive

industrial experience of some of the authors. Building on top of the standard scikit-learn [38] library we trained on the support set of each of the test tasks, with hyperparameter choice following a grid-search and validation procedure as detailed in the supporting code and documentation. Our best performing single-task method is RF, which outperforms kNN in all of our experiments.

Additionally, we also consider a graph neural network (GNN) baseline trained from scratch on each individual task, primarily to illustrate that this is not a promising method. Concretely, we use a GNN with 8 layers, a hidden dimension of 128 and a gated readout function similar to Gilmer et al. [20], referring to it as GNN-ST below. For this, as well as for the other GNN-based models introduced below, we determined hyperparameters after a small search in an author-defined space, considering around ∼30 configurations per model.

**Multi-Task Pretraining**   A commonly successful approach to few-shot learning is to pretrain a model on a selection of related tasks such that a common model "trunk" learns to extract relevant features, and different "heads" on top of this trunk specialize to the set of known tasks. Such an architecture can then be fine-tuned to a new task by "re-heading", i.e., keeping the trained trunk of the model and adding a freshly initialized head for a new task. Fine-tuning then only needs to adapt the parameters of this new head. This idea has been applied to a number of molecular activity tasks, e.g., by Hu et al. [23].

Here, we represent this approach using a GNN-based multi-task model (called GNN-MT below). Concretely, we use a shared GNN model of 10 layers, hidden dimension 128 and using principal neighborhood message aggregation [14] for all tasks, and then have task-specific gated graph readout functions (as Gilmer et al. [20]) and a task-specific MLP with one hidden layer of dimension 512 to produce an activity label. We train the model on the support sets of all tasks in $\mathcal{D}_{train}$ over multiple epochs. We employ an early stopping criterion based on the suitability for specialization to a new task. More precisely, to evaluate the model during training, we iterate over the tasks in our validation set, and initiate a fine-tuning training process on each task's support set, starting from the pre-trained shared GNN and a freshly initialized graph readout layer and final MLP. For each validation task, we record the $\Delta$AUPRC after fine-tuning, and stop training of the model once the mean of these values has stopped improving. To evaluate the trained model, we follow the same fine-tuning strategy for the tasks in our test set.

**Self-Supervised Pretraining**   After the success of self-supervised methods such as BERT [16], similar methods have been developed for use on molecular data. In particular, Hu et al. [23] introduced the idea of pretraining GNNs by masking and reconstructing of node features and substructures. Maziarka et al. [32] implements a similar idea in the *Molecule Attention Transformer* using a Transformer architecture [57] and reports substantially stronger results. We refer to this as MAT below and treat it as a representative of self-supervised methods, but point out that this area is very active and more recent methods may perform even better. To test it on our dataset, we use the released pre-trained model and code, and fine-tune it on the support set of the test tasks (splitting out 20% of the per-task support set as a validation set to allow training with early stopping). We performed a small hyperparameter search on the validation tasks to identify the learning rate that showed best results in fine-tuning.

**Meta-Learning Methods**   Meta-learning methods are designed specifically for the few-shot learning setting, aiming to "learn how to learn". In particular, they learn models that are suitable for rapid adaptation to a new task with very small support sets. Commonly, they take an episodic approach to pretraining: in each episode $e$, a batch of tasks is sampled uniformly $\{\mathcal{T}_k\}_{k=1}^{N_e}$ from $\mathcal{D}_{train}$ and used to train the meta-learner [18]. We choose two key methods as examples here: *optimization-based* meta-learners adapt by taking gradient steps on the support set of a new task [18, 39] and are then used to classify final query examples; *metric-based* approaches compute per-class embeddings of the support set, and classify query examples according to their distance from each [47, 58].

Concretely, we consider Model-Agnostic Meta-Learning [18] procedure on top of a GNN operating on the molecular graph as a representative optimization-based approach and refer to it as GNN-MAML. For this, we use the same GNN as in the GNN-ST case, aiming in particular to evaluate if GNN-MAML succeeds in learning a GNN model that can be rapidly adapted. We use the same procedure to evaluate and fine-tune GNN-MAML as we use for the GNN-MT model.

As a metric-based approach we apply a prototypical network [47] operating on ECFP fingerprints and features extracted by a GNN consuming the molecular graph. We use Mahalanobis distance to measure how similar a new query sample is to representations of the inactive/active classes in the support set of each task, and refer to this model as PN below.

## 6.2 Empirical Results on FS-Mol

We present the results of the discussed baseline methods on FS-Mol in Figure 2. A complete set of results for each task in $\mathcal{D}_{test}$ is available as supplemental material with our source code release at `https://github.com/microsoft/FS-Mol`.

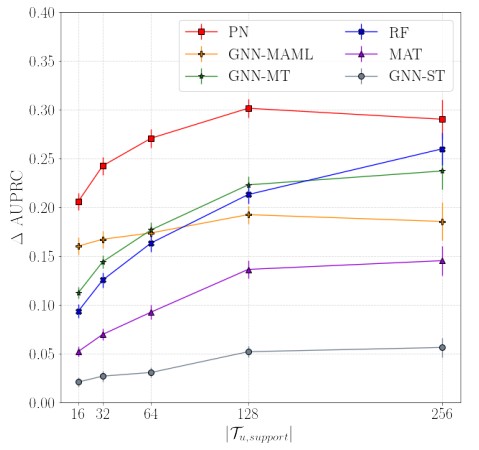
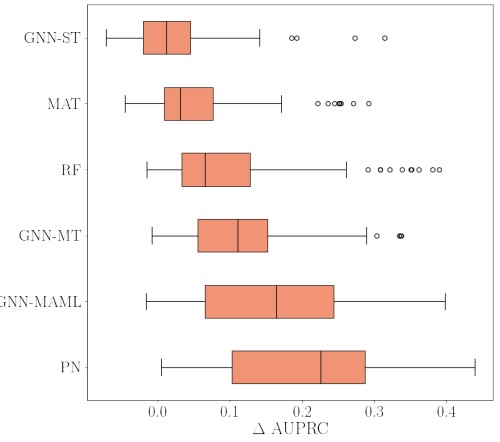

(a) Mean performance on unseen tasks $\mathcal{T}_u$ as the support set size available for adaptation is increased. We include errors in the means for each point.

(b) Performance across all independent unseen tasks from $\mathcal{D}_{test}$, at support set size 16. The boxes represent the interquartile range across tasks, the extended lines are the (5, 95) percentiles and additional points represent outliers.

Figure 2: Experimental baseline results on all few-shot testing tasks from $\mathcal{D}_{test}$

In Figure 2a, we show how performance varies with the size of the used support set as an aggregation across tasks. Regarding (**RQ1**), the results indicate that while GNN-MAML is able to provide gains when only very little training data is present, the random forests often used in real-world applications are doing very well for small-to-medium-sized training sets. However, PN provides a significant performance increase over RF methods, in particular on tasks with few datapoints. Other pre-trained methods (GNN-MT and MAT) either perform comparably to the single-task baseline or under-perform it substantially. The plot also answers (**RQ2**), showing that availability of more training data specific to the task can lead to dramatic improvements for most methods. However, for our meta-learning model GNN-MAML, leveraging information from additional datapoints seems to be challenging, pointing to an interesting question for future research. We note that as $|\mathcal{T}_{u,support}|$ increases, some tasks from $\mathcal{D}_{test}$ can no longer be included as insufficient datapoints are available; the resulting smaller pool of few-shot testing tasks happen to derive a smaller benefit from both the meta-trained and multitask models in this case. This leads to the slight decrease in performance on the aggregation over tasks at $|\mathcal{T}_{u,support}| = 256$.

The results also demonstrate that GNN-ST is clearly outperformed by its pre-trained variants GNN-MT and GNN-MAML (**RQ1**). Similarly, MAT, which uses self-supervised pretraining unrelated to specific tasks, shows some improvement over GNN-ST, but underperforms the GNN-MT, GNN-MAML, and PN models that were pre-trained using task-specific information. The detailed results for support set size 16 in Figure 2b further support these observations, indicating that PN substantially outperforms all other methods in this scenario. However, we stress that performance is highly task-dependent, as evidenced by the broad range of improvements over the full set of few-shot testing tasks in Figure 2b. While in certain tasks, few-shot learning methods clearly outperform single-task, this is sometimes not the case. The degree to which information transfer can occur to a specific new task in few-shot learning is an open question [5, 8].

Table 2: Baseline results on few-shot learning methods on FS-Mol with support set size 16. Results are broken down by few-shot testing class. The reported metric is $\Delta$AUPRC, accounting for the variation in percentage of active compounds in each few-shot testing task. The mean rank is calculated by *autorank* [22], following [15]. Errors in the mean are reported across aggregate task categories, errors across multiple support set splits are present for classes containing single tasks.

| few-shot task categories | | | $\Delta$AUPRC | | | | | |
|---|---|---|---|---|---|---|---|---|
| Class | Description | #tasks | RF | GNN-ST | GNN-MT | MAT | GNN-MAML | PN |
| 1 | oxidoreductases | 7 | 0.081±0.032 | 0.013±0.019 | 0.045±0.013 | 0.062±0.024 | 0.046±0.023 | **0.086±0.029** |
| 2 | kinases | 125 | 0.082±0.006 | 0.013±0.004 | 0.113±0.006 | 0.043±0.005 | 0.178±0.009 | **0.217±0.009** |
| 3 | hydrolases | 20 | 0.158±0.026 | 0.062±0.019 | 0.129±0.025 | 0.095±0.019 | 0.106±0.024 | **0.196±0.031** |
| 4 | lysases | 2 | 0.218±0.172 | 0.161±0.112 | 0.189±0.100 | 0.139±0.105 | 0.218±0.147 | **0.229±0.201** |
| 5 | isomerases | 1 | 0.119±0.029 | -0.014±0.015 | 0.083±0.054 | 0.040±0.044 | 0.006±0.021 | **0.117±0.047** |
| 6 | ligases | 1 | 0.027±0.069 | -0.011±0.003 | 0.046±0.050 | 0.010±0.075 | 0.001±0.017 | **0.058±0.074** |
| 7 | translocases | 1 | **0.102±0.053** | 0.002±0.043 | 0.042±0.031 | 0.067±0.035 | -0.002±0.021 | 0.055±0.021 |
| | all enzymes | 157 | 0.093±0.007 | 0.021±0.005 | 0.112±0.006 | 0.052±0.005 | 0.160±0.009 | **0.206±0.008** |
| | mean rank | | 3.55 | 5.22 | 3.17 | 4.74 | 2.75 | **1.56** |

Table 2 breaks these results up by EC category. In particular, it shows that the largest category of tasks, kinases, which is also most common in the training data (1497 training tasks), clearly profits from meta-learning. Classes 6 and 7, in contrast, are represented in the training set by fewer than 40 tasks each. These results match the intuition that transfer of learned knowledge between more related tasks is more successful [8]. By similar reasoning, we expect the addition of more training tasks to improve few-shot learners. We confirm this empirically in Appendix C, where we show that randomly sub-sampling $\mathcal{D}_{train}$ makes transfer to $\mathcal{D}_{test}$ less effective.

## 7   Discussion & Conclusion

We presented FS-Mol, an up-to-date molecular dataset suitable for evaluation of few-shot learning methods in an important domain outside of computer vision and NLP. In particular, our dataset is chosen to match the modeling task in early-stage drug discovery projects, where the aim is to identify and optimize leads for a specific protein target given very little data. FS-Mol has enabled us to address key research questions. Specifically, it is now possible to evaluate if, and by how much, (new) few-shot methods improve over standard baselines in a realistic QSAR evaluation setting. Our baselines were chosen to be representative of industry practice, so that progress on FS-Mol is more likely to translate into advances in tools for practitioners. Finally, our experiments also allow us to determine the support set sizes for which a few-shot method is likely to be useful, for example showing that MAML is most successful at very small support set sizes.

However, we note that transfer of results to realistic projects is not guaranteed to be successful. In particular, it may be the case that current drug-discovery projects are sufficiently novel such that transferring knowledge from the public ChEMBL corpus does not provide any benefits. This may be alleviated by pre-training few-shot models on more recent, proprietary data that better matches current industry projects. We note that it is an ongoing area of research to understand when transfer-learning is likely to assist on an unseen task [8]. In addition, the few-shot baselines we provide checkpoints and results for are only a representative set, rather than a complete survey of the current state of the field, and so variations of the methods chosen by us may already provide significant boosts.

We are releasing FS-Mol, along with the preprocessing pipeline used to produce it and an extensible implementation of key baseline approaches, illustrating how to benchmark new methods. We hope that the dataset and the implementation will drive further research in few-shot learning, with a focus towards methods that are useful in this important application domain.

**Societal Impact**   While the automation of components of the drug-discovery process could lead to redundancies in the pharmaceutical industry, better QSAR models alone are unlikely to have a large impact, as the tools developed by using this dataset are considerably more likely to be assistive tools for, rather than replacements of, professionals. Finally, we note the potential positive impact of reducing the time and cost required bring a drug to market and the potential of developing more effective treatments using large-scale computational methods rather than the current labour-intensive design-synthesize-measure cycle.

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
