# OpenReview forum: "FS-Mol: A Few-Shot Learning Dataset of Molecules"
_NeurIPS.cc/2021/Track/Datasets_and_Benchmarks/Round2 — NeurIPS 2021 Datasets and Benchmarks Track (Round 2)_

### Official Review · Reviewer_8Tey · 2021-09-13
**Well-curated dataset and benchmarks for a high-impact application area for few-shot learning**

**Rating:** 7
**Confidence:** 3

**Strengths:**

The dataset provided by this paper is useful to both the few-shot learning and ML in chemistry communities.
* The dataset helps broadens the scope of few-shot learning beyond the fields where it is most established. The problem addressed is well-formulated, of clear importance, and involves an important but relatively less-explored data modality (graphs).
* Much recent work in deep learning for chemistry has focused on the large-data regime; such methods are rarely useful for practical problems where data is scarce. This paper helps draw attention to problems of greater relevance/impact to industry practitioners.
* Although the ChEMBL dataset is pre-existing, the authors collect, discretize, and split it in thoughtful manner which helps researchers ask high-level, interesting questions (RQ2, sub-split by EC, etc.) about the dataset.
* The baselines chosen by the authors are appropriate and interesting, spanning from traditional QSAR methods to atomic transformers.
* The results, especially broken down by support set size, are of significant interest to the ML in chemistry community.

**Weaknesses:**

* (RQ2) could be better formulated and explored.
    - (1) The progressive elimination of test-tasks as support size increases leads to artifacts in the results. The authors note this phenomenon in L318; however, ideally the results shown would adjust for this (in one of several possible ways).
    - (2) It would be very useful to see the dependency of few-shot learning methods on the number of train-tasks. I am sympathetic to the fact that it may not be feasible to evaluate all possible combinations for the number of train-tasks, $|\mathcal{T}_\text{support}|$ for train tasks, $|\mathcal{T}_\text{support}|$ for test tasks, etc. However, it would be useful to explicitly mention these variables, and the types of chemical application contexts they may correspond to.

**Additional Feedback:**

Given that some of the EC categories had only one task in $\mathcal{D}_\text{test}$, Table 2 does not add much to the paper. The claim in L330 is also very broad and unconvincing without further elaboration/quantification.

**Clarity:**

The paper is generally well written.
* However, Figures 1(a) and 1(b) could be misinterpreted as plotting cumulative number of compounds vs number of assays. I recommend showing the same data in histogram format, or otherwise emphasize that the assays are ranked by size (of the dataset).
* Details on the model selection / early stopping procedure during fine tuning are lacking.
* Details on the construction of the baselines (L243) are unclear. Are the hyperparameters chosen by the "extensive experience of some of the authors" or the grid search? If the latter (as the text suggests), then what was chosen based on experience?
* A slightly more detailed exposition of the meta-learning methods L284-296 would be helpful as context.

**Correctness:**

* I am concerned by the size-bias in splitting (line 142). We don't get to see, for example, whether including a greater proportion of data-rich tasks in the train-task set would help meta-learning more (or less).
* Another very concerning possibility is that there is some covariate to dataset size that would skew the results. While this cannot be totally eliminated, it would be better to reconstruct train-test splits for each value of $|\mathcal{T}_\text{support}|$

**Documentation:**

Authors provide good documentation of data collection and organization in their GitHub repository. The dataset is also easily accessible and clearly documented there. The models appear easily reproducible.

**Ethics:**

No ethical concerns.

**Relation To Prior Work:**

While the dataset itself is not new, and is also quite similar to many existing datasets, authors make a convincing argument for its relative strengths, mainly in terms of number of tasks. I'm not convinced that the large number of molecules per task in the other datasets is a weakness of the previous datasets, though, as it's always possible to choose $|\mathcal{T}_\text{support}|$ to be as small as necessary. The class balance argument is similarly unconvincing for similar reasons.

**Summary And Contributions:**

This paper frames lead optimization in drug discovery as a few-shot learning problem, and presents a well-selected yet extensive subset of ChEMBL bioassays as an accessible starting point for further development of such methods. The dataset has distributional properties which differ from existing datasets and corresponds to an important stage in drug discovery where modern deep learning methods have yet to had a significant practical impact. The paper provides well-chosen baselines corresponding to classical ML methods, single-task deep learning methods, and deep learning methods leveraging few-shot learning.

---

> ### Author Response · Authors · 2021-09-27
> **Request for clarification -- weaknesses comments**
>
> In the section "weaknesses" you note that "The progressive elimination of test-tasks as support set size increases leads to artifacts in the results. The authors note this phenomenon in L318; however, ideally the results shown would adjust for this (in one of several possible ways). " We will include a plot where we remove the testing tasks which are "too small" to provide complete curves up to and including $|T_{support}| = 256$ but in advance of a more complete response, we request clarification of what might be considered a suitable alternative approach to the manner of plot in Figure 2a.

---

> ### Author Response · Authors · 2021-09-28
> **Author Response**
>
> We thank you for your careful review and detailed, useful feedback.
>
> > (RQ2) could be better formulated and explored.
> > The progressive elimination of test-tasks as support set size increases leads to artifacts in the results. The authors note this phenomenon in L318; however, ideally the results shown would adjust for this (in one of several possible ways).
>
> We request clarification on this point; our suggestion is to update Figure 2a to include only those test tasks with more than 256 points so that the entire $|T_{support}|$ range is always covered, but note that a limitation is that only 42 of the testing tasks are then represented in full.
>
> >  It would be very useful to see the dependency of few-shot learning methods on the number of train-tasks. I am sympathetic to the fact that it may not be feasible to evaluate all possible combinations for the number of train-tasks,  $|T_{support}|$ for train tasks, $|T_{support}|$ for test tasks, etc. However, it would be useful to explicitly mention these variables, and the types of chemical application contexts they may correspond to.
>
> Thank you, we too consider the question of volume of training data an interesting point, see Appendix C (with an updated plot to reflect further datapoints with more training tasks). $|T_{support}|$ for training tasks was taken as a hyperparameter in the search performed prior to evaluation on test tasks. Indeed, we did not evaluate on all testing tasks for every value of $|T_{support}|$ used in the training phase, due to feasibility. However, we were careful to establish the best model using our validation task set ${D_{valid}}$ for each method.
>
> > I am concerned by the size-bias in splitting (Line 142). We don’t get to see, for example, whether including a greater proportion of data-rich tasks in the train-task set would help meta-learning more (or less).
>
> Indeed, this is an important and nuanced point. While we did perform some experiments with early variants of this dataset, this is not studied in detail.
> We note that during training of our few-shot models, each model only uses a sample of each task, with a limit on the number of samples. We found this to be crucial to avoid the model to over-specialize to data-rich tasks, and tuned the size of samples by comparison of results on the validation set of tasks.
>
> > Another very concerning possibility is that there is some covariate to dataset size that would skew the results. While this cannot be totally eliminated, it would be better to construct train-test splits for each value of $|T_{support}|$.
>
> We have requested clarification on this point below.
>
> > Details on the construction of the baselines (L243) are unclear. Are the hyperparameters chosen by the "extensive experience of some of the authors" or the grid search? If the latter (as the text suggests), then what was chosen based on experience?
>
> The hyperparameters for these single tasks methods were chosen based on a grid search, where the space of hyperparameter settings was chosen by the authors who have experience in using these methods in real-world drug discovery projects. A clarifiying addition has been made to the relevant text.
>
> > Details on the model selection / early stopping procedure during fine tuning are lacking.
> > A slightly more detailed exposition of the meta-learning methods L284-296 would be helpful as context.
>
> Both of these points are addressed in a new section of the Appendix: "Meta-learning methods and model selection". Under the space constraints we felt that a more detailed description could not be adequately served in the main body of the text.
>
> > However, Figures 1(a) and 1(b) could be misinterpreted as plotting cumulative number of compounds vs number of assays. I recommend showing the same data in histogram format, or otherwise emphasize that the assays are ranked by size (of the dataset).
>
> A clarification that the assays are ranked by size was added to relevant captions. This choice of plot was made to allow easy comparison with a similar plot used in "Industry-scale application and evaluation of deep learning for drug target prediction", Sturm et al., J Cheminform **12**, 26 (2020).

---

> > ### Author Response · Authors · 2021-09-28
> > **Continued response**
> >
> > > Relation to prior work
> >
> > We accept that other datasets may be subsampled to perform similar function, and it is possible for some component tasks to be sampled to ensure class balance, although many are sufficiently imbalanced that subsampling would be subject to high variance. However, these datasets are not supplied with this in mind, and we have here created a single well-defined set on which consistent comparison can be carried out, complete with a clear methodology. For instance, we note that while three works (“Meta-Learning GNN Initializations for Low-Resource Molecular Property Prediction”, Nguyen eta al. ICML Workshop on Graph Representation Learning and Beyond 2020; “Making graph neural networks worth it for low-data molecular machine learning”, Pappu and Paige, CoRR 2020; “Strategies for Pre-training Graph Neural Networks”, Hu et al., ICLR 2020) have used the LSC dataset for evaluation of few-shot methods in this setting, there can be no direct comparison of these works as a methodology and set of testing tasks is not defined. Therefore, we felt that a single clearly-defined dataset and benchmark has value in this field.
> >
> > > Given that some of the EC categories had only one task in Dtest, Table 2 does not add much to the paper. The claim in L330 is also very broad and unconvincing without further elaboration/quantification.
> >
> > We include the division in to EC categories primarily to indicate that the aggregate results of Figure 2a in particular should not be considered as a complete picture – to reinforce the gains from pretraining are highly task-dependent. While the EC category division in this instance may appear rather arbitrary, it is one way to break down the testing tasks without listing separate results for all 157. We provide complete results for all tasks separately in our github repo.
> >
> > We agree that the claim in L330 is not well supported and have thus changed the wording to better reflect the supported results.
> >
> > We thank you again for your detailed and thorough feedback, it is greatly appreciated.

---

> ### Comment · Reviewer_8Tey · 2021-10-04
> **Reviewer response**
>
> Thank you for the response to the concerns. The discussion of the value of the proposed dataset in the context of existing ones has improved. I apologize for not having a chance to respond to the requests for clarification; however, the concerns there are relatively minor and do not affect the suitability of the paper for acceptance. I am happy to wait for future work to address the detailed nuances of (RQ2).

---

### Official Review · Reviewer_vVK6 · 2021-09-17
**A valuable dataset for few-shot learning in drug discovery**

**Rating:** 8
**Confidence:** 3
**Clarity:** Yes

**Strengths:**

- Standardized datasets and benchmarks for QSAR across a wide range of targets is a useful resource and contribution to the field
- Code and data is easily accessible on Github, along with processing code needed to adapt and generate new datasets, as well as update as the underlying CHEMBL data grows
- Ability to specify alternative featurizations for input molecules is useful and improves accessibility
- Benchmarking results provide useful context and comparison between techniques, particularly as performance varies with different dataset sizes

**Weaknesses:**

- Splitting the tasks into train/test by protein target makes sense, but there may still be highly related/homologous proteins that span between both sets. It would be good to see an assessment of how similar or different the protein topologies are between each of the splits, and how this impact generalization performance
- Also related to splitting, in Lines 138-139 it is unclear to me if the entire test set consists only of enzymes, or if they simply partitioned the subset of the test set by EC number. If the former, it would be good to have a wider distribution of protein families in the test set.
- For the self-supervised baseline, it seems the authors only fine-tuned on the support sets of the test tasks. A stronger baseline would be to fine-tune a self-supervised model as a feature extractor on the training set tasks before evaluating on the test set (similar to the procedure used for the multi-task baseline).

**Additional Feedback:**

None

**Correctness:**

Datasets are well constructed and procedure is fully described in paper and code.

**Documentation:**

Data and code is available and reproducible. There is not a clear maintenance plan; although the code is provided for updating the datasets, it is not clear how often this will be done.

**Relation To Prior Work:**

Yes

**Summary And Contributions:**

This paper presents FS-Mol, a curated dataset for benchmarking few-shot learning methods and evaluating potential of such methods to improve performance on small labeled datasets. The authors generate a large set of tasks consisting of molecules and corresponding biological activity measurements from CHEMBL, and define standardized splits into training, validation, and testing sets. They benchmark several machine learning techniques, including few-shot and meta-learning techniques as well as single-task and self-supervised baselines. Results show that few-shot learning techniques improve over single-task and self-supervised baselines for tasks with small training data, but the benefit decreases as the amount of data for a new task increases.

---

> ### Author Response · Authors · 2021-09-28
> **Author Response**
>
> We thank you for your detailed and thoughtful review, and excellent comments.
>
> > Splitting the tasks into train/test by protein target makes sense, but there may still be highly related/homologous proteins that span between both sets. It would be good to see an assessment of how similar or different the protein topologies are between each of the splits, and how this impact generalization performance.
>
> Thank you, this is an interesting point and one which we spent considerable time in discussion of the best approach. We agree that in general there are likely to be highly related or homologous proteins spanning both sets, despite the care taken to avoid identical target duplication. A challenge in working with the database in general is the precision surrounding available target information; we would have liked to include a detailed analysis of similarities of proteins for all assays, but the contents of free text boxes used to record much of this information are of varying quality. As a result, we chose to focus on enzymes as EC numbers are available. Our division of testing tasks in to categories by EC number is to allow some reflection on this similarity; while the training tasks include proteins with and without EC numbers, the distribution of test tasks for each EC category reflect the distribution of known EC numbers in training. We note that the test tasks reflect this distribution by necessity (due to paucity of data for some classes) but also that performance on the testing tasks broadly reflects this balance; kinases are well represented in training and thus the performance in testing is particularly improved. Interestingly, lysase testing-tasks benefit significantly in spite of what appears to be a relatively small amount of training data -- it is likely that some training tasks where a clear EC  number could not be obtained, or where the protein classification was otherwise unclear, are in fact homologous to some of these lysase test tasks.
>
> We agree a significant improvement and potential extension of this work is a detailed analysis based on protein homology; such an analysis would hopefully enable a user of the dataset to identify whether pretraining is likely to be valuable to their particular protein target. However, this information is more challenging to derive. It is also unclear which representation of a protein is most useful to assess similarity in the context of ligand binding, and we consider this a fascinating avenue of inquiry for future research.
>
> > For the self-supervised baseline, it seems the authors only fine-tuned on the support sets of the test tasks. A stronger baseline would be fine-tune a self-supervised model as a feature extractor on the training set of tasks before evaluating on the test set (similar to the procedure used on the multi-task baseline).
>
> Indeed, fine-tuning on the train set of tasks prior to further fine-tuning on the test set would be expected to provide a stronger baseline. Our current results reflect application of the method as given in the original paper (“Molecule Attention Transformer”, Mazriarka et al., Graph Representation Workshop NeurIPS 2019), where only self-supervised node-level pretraining is applied. We thank you for the suggestion, and will include this baseline in our final version of baseline results.
>
> > Also related to splitting, in Lines 138-139 it is unclear to me if the entire test set consists only of enzymes, or if they simply portioned the subset of the test set by EC number. If the former, it would be good to have a wider distribution of protein families in the test set.
>
> Our entire test set consists of enzymes. We saw this as a trade-off between maintaining diversity across the test set, while being able to make meaningful comment about the similarities between the contents of the training and test sets. Other protein families are present in the training set, but without highly confident protein descriptions or featurization for similarity analysis we were concerned a choice of other tasks might be slightly arbitrary.
>
> We thank you again for your detailed review and note that many of the questions raised are components of ongoing research and follow-up work as we consider them deeply interesting.

---

### Official Review · Reviewer_6DT7 · 2021-09-23
**Good dataset with sensible experiment and validation design**

**Rating:** 7
**Confidence:** 3
**Clarity:** Yes, the paper is well written.

**Strengths:**

1. The crucial parts of the work: dataset preparation, splitting, task design, and benchmarking procedure, are very clearly described.
2. The chosen baselines seem representative and their performance is analysed well with intuitions.
3. The provided code extensive and is well-structured

**Weaknesses:**

Disclaimer: This work falls out of my area of expertise. Judging the work purely as a dataset and benchmarking work, I see very few weaknesses.

1. In Section 5.1 (Evaluation Methodology), the authors choose random splitting. However, does a five-fold stratification suffice to negate the noise due to 'lucky' splits? (In my experience in medical data, it does not...)
2.  "We used typical configurations for these classes of models based on the extensive industrial experience of some of the authors." (Line 243) could be replaced with the lessons that these authors can provide to the community.

**Additional Feedback:**

I really appreciate the attention given to the construction of the dataset and the content of the github repository.

**Correctness:**

Yes. From a FS perspective, the dataset is constructed correctly and documented appropriately.

**Documentation:**

Yes, the documentation and the provided code (with the checkpoints of the baselines) is extensive.

**Ethics:**

As the dataset is constructed from ChEBML, a public dataset, I don't see any ethical concerns.

**Relation To Prior Work:**

Related work in Section 4 and Table 1 is well summarised.

**Summary And Contributions:**

This work provides a molecular dataset for modelling QSAR. Motivated by the real-world settings of limited datapoints and structuring reusability among molecules being experimented with, the work proposes a few-shot learning framework. Augmenting this, the work also provides a benchmarking procedure considering the datapoint per task as well as the number of tasks.

---

> ### Author Response · Authors · 2021-09-28
> **Author Response**
>
> We thank you very much for your careful review and valuable comments.
>
> > In Section 5.1 (Evaluation Methodology), the authors choose random splitting. However, does a five-fold stratification suffice to negate the noise due to 'lucky' splits? (In my experience in medical data, it does not...)
>
> We mostly report aggregate results over the 157 test tasks with five splits each, and hence believe the effect of noise to be relatively small. The choice of five-fold splitting was taken as a trade-off between noise and time-to-evaluate, particularly in the case of methods requiring fine-tuning such as MAML and multitask learning. However, to improve the results for each of the 157 tasks (available in the detailed breakdown table on the github repo) we are reevaluating all of our final models with 10-fold splitting and will update our results and plots accordingly.
>
> > "We used typical configurations for these classes of models based on the extensive industrial experience of some of the authors." (Line 243) could be replaced with the lessons that these authors can provide to the community.
>
> In the accompanying code, we provide the configurations of the single-task models in the file fs_mol/baseline_test.py (Ln 29 – 37). The aim is to provide a reliable hyperparameter space for grid search that was in line with industry practitioners typical choice, rather than optimising carefully for each task, in recognition that a model must be consistent over the course of a project as data is supplied. The specific choices are common for industry practitioners; for instance, fingerprint representations of molecules are sparse and therefore it is deemed important that a search over possible RF depths is relatively unconstrained.

---

> > ### Comment · Reviewer_6DT7 · 2021-09-30
> > **Thank you for the clarifications**
> >
> > Thank you for your response and for updating the manuscript. I stand by my score and propose that this work be accepted.

---

### Official Review · Reviewer_7TrT · 2021-09-23

**Rating:** 6
**Confidence:** 2
**Correctness:** The paper is correct to my knowledge
**Clarity:** The paper is written clearly

**Strengths:**

- Few-shot learning is a relevant problem for early-stage drug discovery, and a standardized benchmark for the problem would be a useful resource. The dataset is carefully constructed.
- The paper reports performance of 6 representative baselines.
- The paper is well-written.

**Weaknesses:**

- My main question is why we only consider a small dataset size for training tasks. For the test task, it makes sense that there is a limited number of molecules since the setting considered is early-stage drug discovery. However, for training tasks, it seems reasonable to have activity data on many more molecules. In fact, to construct the dataset, the authors filter away tasks with lots of data points.
- I'm also wondering how tractable it is to do few-shot transfer, when the identity of the target protein is unknown. With just a few molecules and an unknown protein target, transfer seems challenging.
- The dataset is a curated form of a standard database. Careful curation is important, but the contributions are limited in my opinion.

**Additional Feedback:**

Please see comments above

**Documentation:**

There is detailed documentation in the appendix, and the authors provide a URL for the dataset.

**Ethics:**

Not to my knowledge

**Relation To Prior Work:**

The related work focuses on comparing with other datasets, and should be expanded to include prior work on few-shot learning for drug discovery.

**Summary And Contributions:**

- The paper proposes FS-Mol, which is a few-shot learning dataset of small molecules for QSAR.
- The paper includes a benchmarking procedure on this dataset, and report baseline performance

---

> ### Author Response · Authors · 2021-09-27
> **Author Response**
>
> Thank you for your careful review and valuable comments.
>
> >Why do we only consider a small dataset size for training tasks?
>
> We deliberately removed very large assays as these represent High-Throughput Screens (HTS), which are considered noisy, and of poorer quality in the drug-discovery community. In practice, a large percentage of ChEMBL assays contain few points; we include as much high-quality data as possible. We also consider that it is possible in pretraining to overweight towards one task, allowing a single large task to dominate, contains the possibility of degrading overall performance on a more diverse set of dissimilar test tasks. For this reason, in practice our methods draw equally sized portions of data from each task in the training set.
>
> > How tractable is it to do few-shot transfer when the identity of the target is unknown?
>
> QSAR prediction in drug-discovery is a standard technique that relies on making activity predictions in the absence of detailed protein structure knowledge or use of this knowledge.
> We acknowledge that transfer learning in this setting is challenging, but our baselines indicate that some degree knowledge transfer is possible.  Prior works (see for example “Meta-Learning GNN Initializations for Low-Resource Molecular Property Prediction”, Nguyen eta al. ICML Workshop on Graph Representation Learning and Beyond 2020) have also shown some advantage to transfer learning in this setting. However, we note that consistent comparison between works such as this is challenging, which motivated this dataset and benchmark.
> Few-shot learning methods are likely learning a representation of molecules most useful to similarity identification in the context of specific features necessary for protein-binding. Training on large, unrelated, supervised tasks has been seen to be less valuable for this application (see for example “Strategies for Pre-training Graph Neural Networks”, Hu et al., ICLR 2020; in particular Table 1, in which “supervised graph-level pre-training” [with multitask molecular property prediction] yields the most substantial gains).
>
>
> >The dataset is a curated form of a standard database. Careful curation is important, but contributions are limited in my opinion.
>
> The work was in part motivated by the lack of a clear and consistent dataset and benchmark on which to evaluate few-shot methods; prior work had focused on using “LSC” (see “Related Work”) -- a dataset also derived from an earlier version of ChEMBL. However, in the two known works (Nguyen et al., and “Making graph neural networks worth it for low-data molecular machine learning”, Pappu and Paige, CoRR 2020) that address specific few-shot methods, the tasks used to evaluate final results following pretraining differ and no comparison was offered. We also noted inconsistencies in the numbers of molecules reported per assay and those available in the dataset in one of the publications, likely due to the varying possible methods of obtaining the dataset for use. With these challenges, a clear comparison was not easy between the works, or with new techniques. The dataset and benchmark aims to enable consistent comparison of novel methods in QSAR among different authors, and extends the number of training tasks used beyond that in the aforementioned works by a factor > 4.
>
> > The related work focuses on comparing with other datasets, and should be expanded to include prior work on few-shot learning for drug discovery.
>
> We have referenced prior work in this direction in the introduction to the manuscript (Ln 42-45). We emphasised datasets and benchmarks in the related work section as we present this work as a dataset/benchmarking contribution. We have included more detailed discussion of previous work in few-shot learning for molecules in our extended “Related Work” section in Appendix B.

---

### Decision · Program_Chairs · 2021-10-10

**Decision:**

Accept

**Comment:**

This paper proposes FS-Mol, a molecular dataset and benchmarking system especially motivated by recent progress and work on few-shot learning approaches in the biological/molecular domain. Overall, reviewers found this to be a strong and valuable contribution to the community.